# Comparative Study of Toxic Effects and Pathophysiology of Envenomations Induced by *Carybdea brevipedalia* (Cnidaria: Cubozoa) and *Nemopilema nomurai* (Cnidaria: Scyphozoa) Jellyfish Venoms

**DOI:** 10.3390/toxins14120831

**Published:** 2022-11-28

**Authors:** Du Hyeon Hwang, Phil-Ok Koh, Ramachandran Loganathan Mohan Prakash, Jinho Chae, Changkeun Kang, Euikyung Kim

**Affiliations:** 1Department of Pharmacology and Toxicology, College of Veterinary Medicine, Gyeongsang National University, Jinju 52828, Republic of Korea; 2Institute of Animal Medicine, Gyeongsang National University, Jinju 52828, Republic of Korea; 3Department of Anatomy, College of Veterinary Medicine, Gyeongsang National University, Jinju 52828, Republic of Korea; 4Marine Environmental Research and Information Laboratory, Gunpo 5850, Republic of Korea

**Keywords:** comparative study, Cubozoa, Scyphozoa, jellyfish venom extracts, toxicity, hemorrhage, pathophysiology

## Abstract

Jellyfish stings can result in local tissue damage and systemic pathophysiological sequelae. Despite constant occurrences of jellyfish stings in oceans throughout the world, the toxinological assessment of these jellyfish envenomations has not been adequately reported in quantitative as well as in qualitative measurements. Herein, we have examined and compared the in vivo toxic effects and pathophysiologic alterations using experimental animal models for two representative stinging jellyfish classes, i.e., Cubozoa and Scyphozoa. For this study, mice were administered with venom extracts of either *Carybdea brevipedalia* (Cnidaria: Cubozoa) or *Nemopilema nomurai* (Cnidaria: Scyphozoa). From the intraperitoneal (IP) administration study, the median lethal doses leading to the deaths of mice 24 h post-treatment after (LD_50_) for *C. brevipedalia* venom (CbV) and *N. nomurai* venom (NnV) were 0.905 and 4.4697 mg/kg, respectively. The acute toxicity (i.e., lethality) of CbV was much higher with a significantly accelerated time to death value compared with those of NnV. The edematogenic activity induced by CbV was considerably (83.57/25 = 3.343-fold) greater than NnV. For the evaluation of their dermal toxicities, the epidermis, dermis, subcutaneous tissues, and skeletal muscles were evaluated toxinologically/histopathologically following the intradermal administration of the venoms. The minimal hemorrhagic doses (MHD) of the venoms were found to be 55.6 and 83.4 μg/mouse for CbV and NnV, respectively. Furthermore, the CbV injection resulted in extensive alterations of mouse dermal tissues, including severe edema, and hemorrhagic/necrotic lesions, with the minimum necrotizing dose (MND) of 95.42 µg/kg body weight. The skin damaging effects of CbV appeared to be considerably greater, compared with those of NnV (MND = 177.99 µg/kg). The present results indicate that the toxicities and pathophysiologic effects of jellyfish venom extracts may vary from species to species. As predicted from the previous reports on these jellyfish envenomations, the crude venom extracts of *C. brevipedalia* exhibit much more potent toxicity than that of *N. nomurai* in the present study. These observations may contribute to our understanding of the toxicities of jellyfish venoms, as well as their mode of toxinological actions, which might be helpful for establishing the therapeutic strategies of jellyfish stings.

## 1. Introduction

Jellyfish stings are a great public health concern in many parts of the world [1,2]. Over the last several decades, there have been huge expansions of jellyfish blooms [3] worldwide, with an estimated 150 million jellyfish stings annually [4]. Sometimes, the jellyfish stings are accompanied by life-threatening complications, such as diffused neurotoxicity, cardiovascular collapse, respiratory failure, hypotension, shock, and even death [5,6].

Jellyfish belong to the phylum Cnidaria which has five classes, including Anthozoa, Cubozoa, Hydrozoa, Scyphozoa, and Staurozoa [7,8]. These Cnidarians have a common feature, namely a large number of stinging organelles (i.e., nematocysts). Nematocysts are used for defense, prey capture, and spatial competition, and following stimulation, the nematocysts inject venomous substances into the integument of the prey or predator [9]. Nematocysts contain highly complex and toxic mixtures of proteins, peptides, and small molecules which are the underlying basis of severe envenomations induced by jellyfish stings. It has also been proposed that there can be large variations in jellyfish venom compositions, along with their toxicities and the associated symptoms and signs depending on the species [10,11]. Jellyfish envenomation can manifest as local tissue effects as well as fatal systemic reactions [12,13]. Among these, cutaneous tissue damage is most frequently observed in jellyfish stings. Local tissue damage can culminate in disability or complications if left untreated or upon delay of the administration of drugs and proper management. Current treatment approaches to manage skin lesions, including rash, itching, and oedema include the use of oral and topical antihistamines [14,15]. Unfortunately, antivenom therapy for neutralizing jellyfish venom is barely explored. For the development of a therapeutic strategy for jellyfish stings, it is important, not only to characterize the modes of action but also to objectify the degree of envenomation, such as the quantification of their toxic effects.

The aims of the present study are to quantify and compare the toxic effects of the major venomous species in Korean waters, which are *Carybdea brevipedalia* (Cnidaria: Cubozoa) and *Nemopilema nomurai* (Cnidaria: Scyphozoans). In general, box jellyfish (Cnidaria: Cubozoa) are considered the world’s most venomous animals, which can cause medically important stings [16]. Especially, the stings of a Cubozoa, *Chironex fleckeri,* [17,18], can cause not only severe pain and local tissue damage but also occasional cardiovascular arrest and even to death [17]. One of the Cubozoa species, *C. brevipedalia* (named *C. mora)* [19,20] is widely distributed in the waters of the Pacific Ocean, including Australia, California, Hawaii, the Philippines, Japan, and Korea [21]. They produce immediate pain, skin lesions in the form of vesicles, swelling, red papules, and inflammation with red marks. These victims may also suffer from symptoms like cardiac and/or neurological complications [22,23,24,25,26]. The giant jellyfish *N. nomurai* is a Scyphozoa (true jellyfish) and another venomous species that mainly blooms along the East Asian marginal seas, principally distributed along the coasts of China, Korea, and Japan [27]. Although the venom is not as toxic as that of Cubozoans, envenomations by these jellyfish are common and induce mild to moderate topical symptoms including redness, oedema, itching, immediate pain, and inflammation. It can also be accompanied by other types of toxinological symptoms, such as cytotoxicity, hemolysis, cardiotoxicity, and even death [10,28].

Currently, to our understanding, there are few standardized measurements to estimate the jellyfish venom extract toxicities of diverse origins using objective and quantitative methodologies. In doing so, we can more accurately evaluate the toxicities of venoms as well as characterize the mechanisms of their toxic actions. For this, we are driven to establish a standardized measurement for determining the toxicities of two venomous jellyfish species (*C. brevipedalia* and *N. nomurai*). These results may confer a guide for the development of an effective therapeutic strategy for venomous jellyfish stings.

## 2. Results

### 2.1. Jellyfish Collection and Venom Extraction

Jellyfish specimens of two different venomous species were collected in August of 2019 from the coasts of South Korea as follows, *C. brevipedalia* and *N. nomurai* from the Korea Strait near Samcheonpo and Geoje-do, respectively. Stations from which two different venomous species of jellyfish were collected are marked by dots in Figure 1. Crude venoms of *C. brevipedalia* (CbV) and *N. nomurai* (NnV) were extracted from batches of cleaned nematocysts as shown in Figure 2B,D. Nematocysts mixtures for jellyfish venom extraction were checked microscopically.

### 2.2. Comparative Lethality of Crude Jellyfish Venoms from Cubozoan and Scyphozoan Classes

We have compared the lethality and various toxinological effects of venoms between the two representative Cnidarian classes (*C. brevipedalia* as a Cubozoa, and *N. nomurai* as a Scyphozoa). The lethality (as LD_50_) of CbV and NnV was determined using mice by intraperitoneal (i.p.) injection of the venoms (0.25–20 mg/kg) (Table 1). Although the venom treatments have increased the lethality of mice in a dose-dependent manner (Figure 3), there was a large difference in their potencies between the two jellyfish species. The CbV (LD_50_ = 0.9052 mg/kg) was much more potent than NnV (LD_50_ = 4.4697 mg/kg). The relative potency ratio (CbV versus NnV) of LD_50_ values indicates CbV has a 4.938-fold higher potency than NnV. From this experiment, the highest lethality (LD_100_) was observed at 5 and 20 mg/kg for CbV and NnV, respectively. The relative potency ratio (CbV versus NnV) of LD_100_ suggests CbV has a 4.0-fold higher potency than NnV.

For further analyses of venom potency, the plots of CbV and NnV dose versus time to death had been determined and they showed a curvilinear relationship. The latency of death induction in mice showed a significant negative correlation with the administered dose, meaning high doses lead to faster and stronger lethality than lower doses. At the highest dose (20 mg/kg) of CbV, the mice died at 1.84 ± 0.38 h, whereas it was 5.21 ± 1.53 h with NnV (Figure 3), indicating that CbV is more potent than NnV. Considering the time requirements for inducing lethality, CbV resulted in mice death 2.831-fold (=5.21/1.84) more quickly than NnV. In addition, the calculated doses leading to death in mice within 10 h of the treatments (we call it LD_10h_) for CbV and NnV were 1.22 mg/kg and 6.25 mg/kg, respectively. CbV demonstrated a stronger lethality and significantly accelerated the time of death compared with NnV. The relative potency (CbV versus NnV) of LD_10h_ suggests that CbV has a 5.123-fold (=6.25/1.22) higher lethality than NnV.

### 2.3. Edematogenic Activity of Jellyfish Venoms

The potential edematogenic activities of the venoms were estimated in mice and compared the difference (Figure 4). BALB/c mice were treated with a subplantar injection of various concentrations of venoms into the right hind paw of the animals, and their swollen paws were measured at different time points. As shown in Figure 4A, the mice paw edema could be observed from the injected sites in a venom dose-dependent manner. Approximately 1 h after the administration, the venoms (100 μg of venom/paw) of CbV and NnV caused marked increases of hind paw edema, with the maximal effects of 183.57% and 125% compared with the vehicle (PBS) control, respectively. From this, the edematogenic activity induced by CbV was significantly (83.57/25 = 3.343-fold) greater than NnV. In addition, the paw edema induced by the venoms was long lasting, up to 24 h. The edema was accompanied by an apparent hematoma in the mice administered with CbV, but not with NnV (Figure 4B). The mice paw that received PBS alone showed a slight increase in paw size within 0.5–1 h, then the paw size resolved back to normal within 3 h after administration. Further, the histopathological images of paw tissues in CbV- or NnV-treated groups revealed an increase of edema and neutrophil infiltration, in comparison with the control group (Figure 4C). Of further note, the skin thickenings of both epidermal and dermal tissues were noticeably increased in the CbV-treated mice. These results indicate that there was severe subcutaneous edema and marked increases in inflammatory cells, and neutrophil infiltration by the venom treatment. Furthermore, thrombosis-like changes could be also observed in many blood vessels of the deep dermis in the CbV-treated mice.

### 2.4. Hemorrhagic Activity of Jellyfish Venoms

The jellyfish venoms were not only edematogenic but also caused hemorrhage and a subsequent progression to skin necrosis. The onset of a jellyfish venom-induced hemorrhage appears to be due to the degradation of the extracellular matrix (ECM) and basement membrane proteins surrounding the blood vessels by toxic proteases. To investigate the hemorrhagic activity of CbV and NnV, the venoms were intradermally injected into mice’s dorsal skin with various concentrations (Figure 5A). Macroscopic analysis of the inner surface of injected skins showed hemorrhagic changes (Figure 5B). After 3 h of intradermal injection, the mice were euthanized, and the skins were removed for examination. The minimal hemorrhagic dose (MHD) was applied to estimate the hemorrhagic potency. The MHD of the venoms were found to be 55.6 and 83.4 μg/mouse for CbV and NnV, respectively. CbV showed more severe hemorrhagic changes in the subcutaneous tissue, whereas less severe hemorrhagic changes were identified from NnV, and no significant alteration was detected in the control mice’s skin.

### 2.5. Skin Necrotic Activity of Jellyfish Venoms

Intradermal injections of either CbV or NnV resulted in damage to the mice’s skin tissue irritating skin necrosis (Figure 6). The lesions of dermal necrosis were assessed from the inner side of the mice’s skin at 72 h after venom administration. Both CbV and NnV treatments showed significant dose-dependent increases in dermonecrotic lesions (Figure 6A,B). The results show an MND of 95.42 and 177.99 μg for CbV and NnV, respectively (Figure 6B). We evaluated the histopathological alterations of dermonecrotic skin lesions induced by either CbV or NnV (Figure 6C). From this, the control group showed neither any histopathological alteration nor any infiltration of inflammatory cells. Whereas CbV treatment caused substantial dose-dependent pathological changes in the epidermis which had been found to be even totally lost in some areas, the necrotic tissues revealed cellular debris in the form of numerous rounded collections or diffuse infiltration, constituting a necrotic plaque focally detached from the deep tissue. The skin tissues were evaluated for the individual layers, namely the epidermis, dermis, hypodermis, and muscle (Figure 6D). The severity of skin necrosis was classified as normal (score of 0), minimal (score of 1), mild (score of 2), moderate (score of 3), or severe (score of 4) (Table 2). The necrosis of skin tissues was intense, affecting the skeletal muscle and presenting cellular debris infiltration. The NnV treatment group presented dermal necrosis and mononuclear cell infiltration in the dermis and subcutaneous layers. However, it was not as bad as that of CbV treatment. Further analyses using the TUNEL assay demonstrated that the treatments of mouse skin with either CbV or NnV resulted in dose-dependent apoptotic cell deaths (Figure 6E,F). Skin necrotic changes and damages were much more prominent with CbV in comparison with those with NnV.

## 3. Discussion

Jellyfish stings cause toxic or lethal responses in the animal kingdom. During the last several decades, an increase in jellyfish blooms have been observed in many places around the world. They have often come into accidental contact with humans, interfering with human activities as well as imposing a considerable impact on public health, safety, and socioeconomy [29,30,31,32]. In most cases, jellyfish stings induce mild to moderate topical symptoms including redness, oedema, itch, and acute pain. These topical cutaneous symptoms are often inflammatory and termed jellyfish contact dermatitis [33]. Moreover, a high percentage of survivors of these jellyfish stings were left with marked tissue damage at the site of venom injection. Jellyfish dermatitis is the result of the combined effects of various toxic materials in the venoms of jellyfish nematocysts [10,28,34].

Jellyfish venom extract is a complex mixture of proteins, peptides, and small molecules affecting target organs either individually or in combination, thereby causing local and/or systemic symptoms [35]. Each jellyfish venom may exhibit its unique toxic components, such as metalloproteinase, phospholipases A2 (PLA2), serine protease inhibitor, C-type lectin, pore-forming protein, neurotoxin, cytolysin, and hemolysin [35,36], showing striking diversities and similarities between species. Venom metalloproteinases are well known to play a key role in hemorrhage, fibrinolysis, apoptosis, blood coagulopathy, pro-inflammatory activity, platelet aggregation inhibition, and suppression of blood serine protease inhibitors [37]. Previously, it has been reported that Scyphozoan jellyfish venom contains large amounts of toxic proteases, including metalloproteinases, which contribute to the overall toxic effects and proteolytic activity of the jellyfish venom [38]. It was also suggested that toxic metalloproteinases have a major role in the pathogenesis of affected skin [39], including edema formation, being associated with their vascular basement membrane degradation reaction [34]. PLA2 is also widely found as a major component in venomous animals, inducing edema formation, potent myotoxicity, neurotoxicity, pathogenesis, hemolysis, and lethality [40,41]. Serine proteases were similarly reported from various venom sources [42,43], causing haemotoxicity, fibrinolysis, platelet aggregation, and edema [44]. C-type lectin is universally observed in many snake venoms, causing hemagglutination, platelet aggregation, edema, renal effects, and death [45,46]. Additionally, hemolysins are often found in the venoms of various kinds of poisonous jellyfish species, exhibiting the lysis of red blood cells and potentially causing a high probability of death. [10,47,48,49,50]. Based on previous reports, the box jellyfish venom has labile, basic cytolysins (42–46 kDa) that are not known to occur in other Cnidarians [51,52,53,54]. These cytolysins have been demonstrated to be potentially lethal and cause pain, inflammation, and necrosis of the skin in experimental animals, as pore-forming toxins [47,50,51,52]. For example, the box jellyfish, *Carybdea alata,* has both hemolytic and lethal qualities [52]. *Chiropsalmus quadrigatus* showed hemolytic activity toward sheep red blood cells with an ED_50_ of 160 ng/mL and lethality to crayfish with an LD_50_ of 80 μg/kg [53]. *Carybdea rastoni* venom was hemolytic and fatal to mice at 20 μg/kg (i.v.) [51].

*Carybdea brevipedalia* (Cnidaria: Cubozoa) and *Nemopilema nomurai* (Cnidaria: Scyphozoa) are apparently the two most common jellyfish species, being responsible for stinging accidents in the waters of the East Asian Sea, especially Korea. Previous to this work, there has not been a study to directly compare the toxic effects and pathophysiologic sequalae of the Cubozoa and Scyphozoa. In the present study, we have demonstrated the comparative toxinological effects of *C. brevipedalia* and *N. nomurai* jellyfish venoms, including their lethalities, edematogenic activities, hemorrhagic effects, and skin necrotic changes, as well as their histopathological alterations. The present work has demonstrated that the LD_50_ of CbV was 0.905 mg/kg, showing a higher lethality in comparison with that of NnV (4.4697 mg/kg). This study evaluated the toxinological effects of two representative toxic jellyfish species (a Cubozoa, *Carybdea brevipedalia,* and a Scyphozoa, *Nemopilema nomurai*) in a systematic comparative approach. Furthermore, we compared the edematogenic and hemorrhagic effects of the venoms occurring at the early stages of envenomation, which are in agreement with other reports [35]. The present study also showed that subplantar injection of CbV in mice caused more significant paw edema and persistent inflammation in comparison with those of NnV. Local tissue edema of venom injected sites can be expected as a common response in jellyfish envenomation. It has also been suggested that a considerable contribution of NnV metalloproteinase-like components to the increased edema in the injected mice [34].

Our present results are largely in accordance with other reports of jellyfish stinging accidents in humans [14,32]. Overall, the toxinological effects of CbV were much more potent than those of NnV. The MHD was found to be 55.6 and 83.4 μg for CbV and NnV, respectively. A macroscopic view of relevant mice skin showed local histopathological manifestations such as hemorrhage involving substantial degradation of extracellular matrix (ECM) at the site of envenomation and led to progressive tissue damage and dermonecrotic lesion. The jellyfish venoms markedly affected all the soft tissues (epidermal, subcutaneous, and skeletal muscle tissue). In a previous study, we reported that jellyfish (Scyphozoans) venoms contain proteolytic enzymes capable of degrading distinct proteins such as gelatin, casein, and fibrinogen [38]. These data suggest that the venom proteases may lead to the degradation of proteins and extracellular matrix components, affecting local damage. The MND was found to be 95.42 and 177.99 μg for CbV and NnV, respectively.

## 4. Conclusions

In the present study, our findings showed the comparative toxic and pathophysiological effects of the nematocyst venoms from CbV and NnV, the two major stinging jellyfish in Korea, using various kinds of mouse models. In addition, these results suggest that the LD_50_, MHD, MND, and edematogenic activities appear to be highly different depending on the jellyfish species. The quantitative measurements of jellyfish venom toxicity along with the characterization can be very helpful for establishing the therapeutic strategy of the envenomed patients. The present study may contribute to the understanding of the toxic effects and treatment of venomous jellyfish stings.

## 5. Materials and Methods

### 5.1. Jellyfish Collection and Preparation

Two kinds of venomous jellyfish species were collected from the coasts of South Korea, which are *C. brevipedalia* and *N. nomurai* from the Korea Strait near Samcheonpo and Geoje, respectively, in August 2019. Only tentacles were dissected and transferred to the laboratory immediately in ice and their nematocysts were isolated using the method described by Bloom [55] with a slight modification. In brief, the tentacle samples were gently rinsed with ice-cold seawater to remove any debris. Then they were placed in three volumes of cold seawater for 24 h with gentle swirling at 4 °C. Thereafter, the supernatant without tentacle was carefully decanted and centrifuged at 1000× *g* for 10 min to harvest the nematocysts. The remaining sediment tentacles were further autolyzed with additional fresh seawater and this autolysis process was repeated for 3–4 days. Lastly, the nematocysts collected were lyophilized (freeze-dried) and stored at −70 °C until further use.

### 5.2. Venom Extraction and Preparation

Jellyfish venoms were extracted from the lyophilized nematocysts powder using the technique described by Lee et al. [56] with a minor modification. Briefly, 70 mg of lyophilized nematocyst powder were added with glass beads (0.5 mm in diameter) and 1 mL of ice-cold phosphate buffered saline (PBS, pH 7.4, 4 °C). These sample mixtures were shaken at 3000 rpm for 30 s, which was repeated ten times with intermittent cooling on ice. The venom extracts were then transferred to a new Eppendorf tube and centrifuged (15,000× *g*) at 4 °C for 30 min. The supernatant was used as jellyfish venom extract for the present study. The protein concentration of the venom was determined by the Bradford method (Bio-Rad, Hercules, CA, USA), with bovine serum albumin (BSA) as a protein standard. The jellyfish venoms were used based on their protein concentrations.

### 5.3. Animals

Six-week-old BALB/c mice (18–20 g) were obtained from Samtako BioKorea (Osan, Korea) and cared for in a Gyeongsang national university laboratory animal research center. The animal room was under standard conditions maintained at 23 ± 2 °C with 12 h light and dark cycles. Food and tap water were supplied ad libitum. The assignment of mice to treatment groups was performed randomly. All experimental procedures were approved by the Committee on the Ethics of Animal Experiments of the Institutional Animal Care and Use Committee of Gyeongsang National University, and the animal study protocol number is GNU-191120-R0061.

### 5.4. Lethal Potency of Jellyfish Venoms (LD_50_)

The median lethal dose (LD_50_) of jellyfish venom was determined according to the World Health Organization (WHO) guidelines based on the method of Meier and Theakston (Meier and Theakston 1986). The LD_50_ value is defined as the amount of venom causing death in 50% of injected mice. The lethalities of each jellyfish venom (CbV and NnV) were evaluated using mice intraperitoneal (i.p.) administration. The BALB/c mice were randomly assigned to experimental groups with five mice per each venom dose (0.25–20 mg/kg in 200 μL of PBS), which covered the entire spectrum of mortality from 0 to 100% in the present study. Control mice were administered with PBS alone. The mice were carefully monitored for death over a 24 h period following the injection to calculate LD_50_ (LD_50_). The dose leading to the deaths of mice within 10 h of treatment (we call it LD_10h_) was calculated from the dose-response (time-to-death) curve [57]. The LD_10h_ was also considered for the assessment of venom toxicities of CbV and NnV.

### 5.5. Edematogenic Activity

The edematogenic activities of testing venoms (CbV and NnV) were measured according to the methods described elsewhere [58,59] with a slight modification. In brief, mice were randomly assigned to groups of five mice each and administered subplantarly 100 μL of various concentrations of venoms (50–100 μg/kg) into their right hind paw, and the contralateral hind paw was injected with PBS (control). Then, the thickness of hind paw edema was measured using micrometer calipers at various time points before (E0) and during the 24 h after administration (Et). The increase in paw volume was expressed in percentage (%), based on the calculation using the formula: (Et − E0)/E0 × 100. Further, the hind paw tissues were processed for histopathological examinations.

### 5.6. Hemorrhagic Activity

The hemorrhagic activities of jellyfish venoms were evaluated in the mouse skin tissues according to the method described by Gowda et al. [60] with modification. Experimental animals were randomly divided into groups (*n* = 5). The shaved dorsal skin of mice was intradermally injected with various concentrations (25–200 μg/kg) of venoms in 100 μL PBS. Control mice were injected with 100 μL of PBS instead of venom. After 3 h, the mice were euthanized, the dorsal skin was removed, and the inner surface was evaluated for hemorrhagic activity. The minimum hemorrhagic dose (MHD) was defined as the concentration of venom required to induce a hemorrhagic spot of 1 cm^2^ diameter from the spot of injection. The areas of the hemorrhagic lesions were measured using the formula as follows, Area (A) = Length (L) × Width (W), and analyzed by NIH Image software (Image J). Further, the skin tissues were processed for histopathological examinations.

### 5.7. Skin Necrotic Activity and Score

Skin necrotic activity was evaluated using the method of ref. [61] with a minor modification. Briefly, randomized groups (*n* = 5 per group) of BALB/c mice were intradermally administered 100 μL of PBS having various doses of the venoms (0–200 μg/kg) into the shaved dorsal skin. Control groups were injected with 100 μL of PBS alone. After 72 h, the mice were euthanized and the dorsal skins of the injection sites were removed, and the necrotic lesion diameters were assessed on the inner side of the skins. The areas of the skin necrotic lesions were measured using the formula, Area (A) = Length (L) × Width (W), and NIH Image software (Image J). The minimum necrotic dose (MND) for the present study was defined as the least amount of venom (µg dry weight) which when injected intradermally into mice, induced necrosis of 1 cm^2^ at 3 days following injection. The skin tissues were processed for examining 8+ further histopathological alterations, then the tissue dermal necrosis was scored according to Ho et al. [62] (Table 2).

### 5.8. Histopathological Analyses and Dermal Necrosis Score

The hind paw and skin tissue samples were fixed using a 10% formaldehyde solution for 24 h. After fixing, the paraffin-embedded tissues were serially cut into 5 μm thick sections. Representative sections were stained with hematoxylin and eosin (H & E) and evaluated for any histopathological alterations by light microscope at 100× and/or 200× magnification. (Olympus BX-51; Tokyo, Japan). Then, tissue dermal necrosis was scored according to the scoring system (Table 2) [62].

### 5.9. Terminal Deoxynucleotidyl Transferase-Mediated UTP End Labeling (TUNEL) Assay for Tissue Analysis

The samples were dehydrated, embedded, and performed using the ApopTag^®^ Plus Peroxidase in Situ Apoptosis Kit (Cat # S7101; Millipore, Danvers, MA, USA) following the manufacturer’s protocols with modifications. Briefly, sections were dewaxed in xylene followed by rehydration in a descending concentration of ethanol for 15 min each at room temperature followed by a 5 min PBS wash. After rehydration, sections were permeabilized with proteinase K (1 μg/mL) for 1 min at room temperature and washed with PBS. Endogenous peroxidase was blocked by 3% H_2_O_2_ at room temperature for 10 min and then washed with PBS. After washing, the sections were immersed in a terminal deoxynucleotidyl transferase (TdT) buffer containing deoxynucleotidyl transferase and biotinylated dUTP, and incubated in a humid chamber at 37 °C for 90 min. The TdT enzyme reaction was terminated after 1 h by treating sections with stop buffer included in the kit. After washing with PBS, peroxidase-conjugated antidigoxigenin antibody was added to sections and incubated in a humidified chamber for 30 min at room temperature. After staining with the peroxidase chromogenic substrate 3,3′ diaminobenzidine (DAB), the slides were examined under 100× and/or 200× magnifications using a light microscope (Olympus BX51 microscope; Tokyo, Japan). The brown TUNEL positive cells were calculated after their quantification in 10 randomly selected fields at 400× magnification under the light microscope.

### 5.10. Statistical Analysis

The results are expressed as a mean ± standard deviation (S.D.). A paired Student’s *t*-test was used to assess the significance of differences between the two mean values. * *p* < 0.05 and ** *p* < 0.01 were considered to be statistically significant.

## Figures and Tables

**Figure 1 toxins-14-00831-f001:**
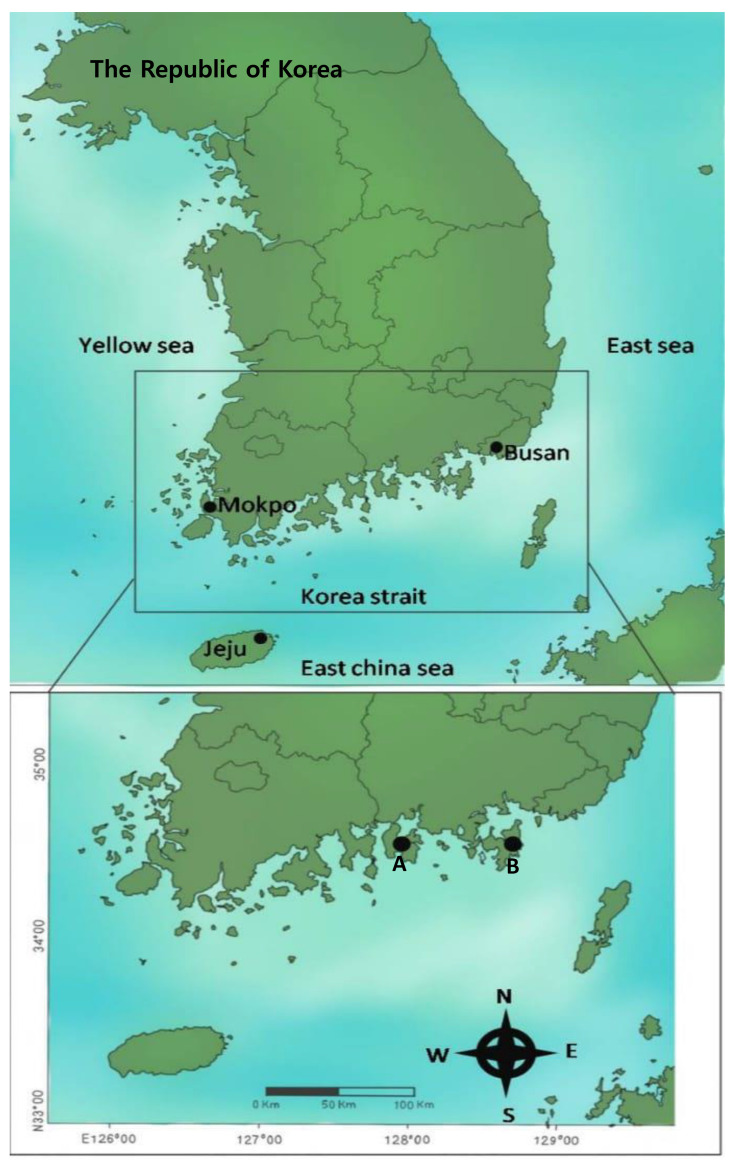
Map of the sampling sites of two venomous jellyfish species. Map-location symbols indicate the sampling localities of the two different jellyfish species (A: *C. brevipedalia* from Namildae in Samcheonpo, and B: *N. nomurai* from Guyeong Beach in Geoje-do).

**Figure 2 toxins-14-00831-f002:**
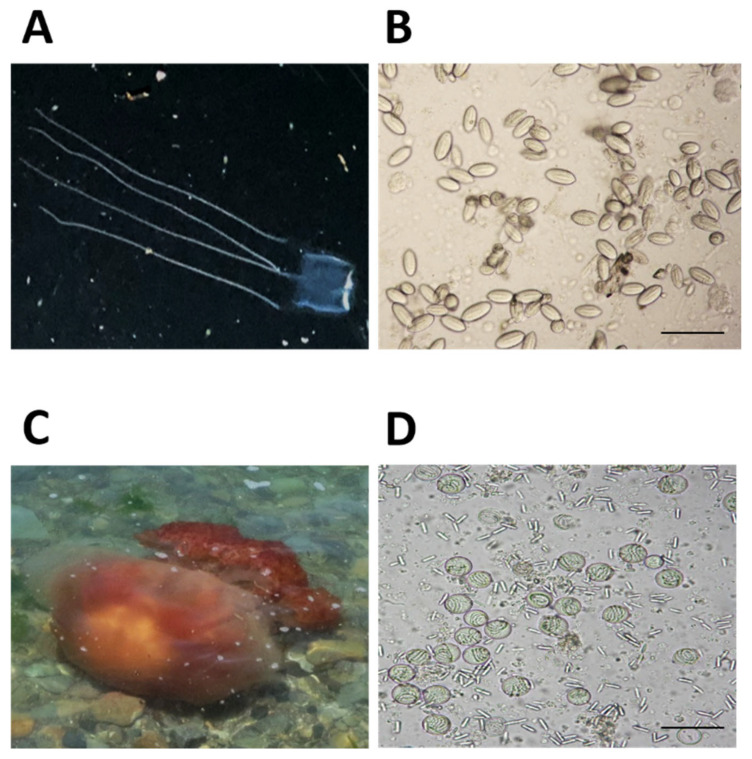
Morphology of *C. brevipedalia* (**A**) and *N. nomurai* (**C**) and their nematocysts (**B**,**D**). (**A**) Mature *C. brevipedalia* major taxonomic characteristics that have four long and fine tentacles. (**B**) A phase contrast microscopic view (100× magnification) of nematocysts isolated from *C. brevipedalia* which was used for the extraction of *C. brevipedalia* venom (CbV). (**C**) Mature *N. nomurai* has major taxonomic characteristics and have grey with pinkish-brown tentacles. (**D**) A phase contrast microscopic view (100× magnification) of nematocysts isolated from *N. nomurai* which was used for the extraction of *N. nomurai* venom (NnV).

**Figure 3 toxins-14-00831-f003:**
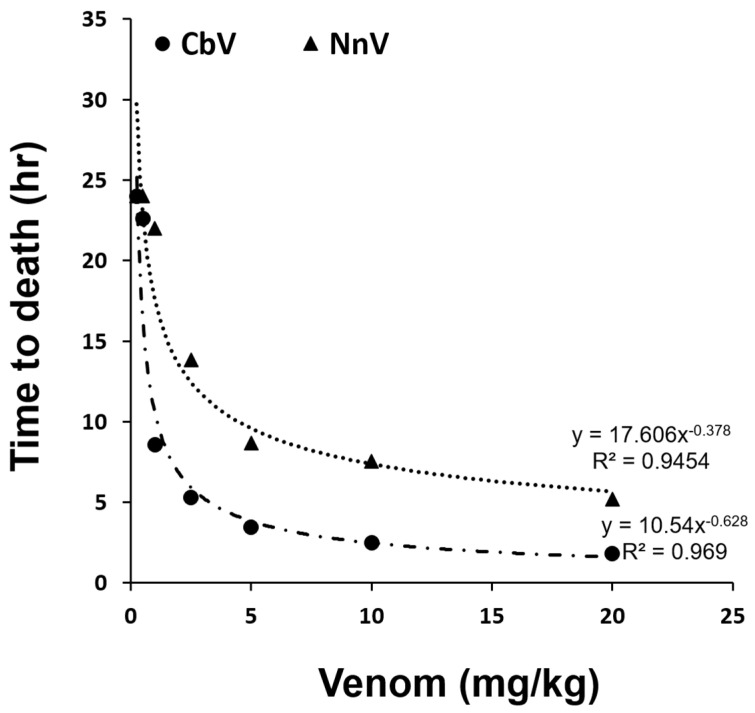
Comparison of the lethalities of CbV and NnV. Dose-response (time-to-death) relationships after either CbV or NnV injections into mice. The curvilinear relationship graphs were plotted to estimate the LD_10h_ for CbV and NnV. The lethality was determined from mice intraperitoneally injected with various venom doses (0.25–20 mg/kg). The values represent the means ± STDEV (*n* = 5).

**Figure 4 toxins-14-00831-f004:**
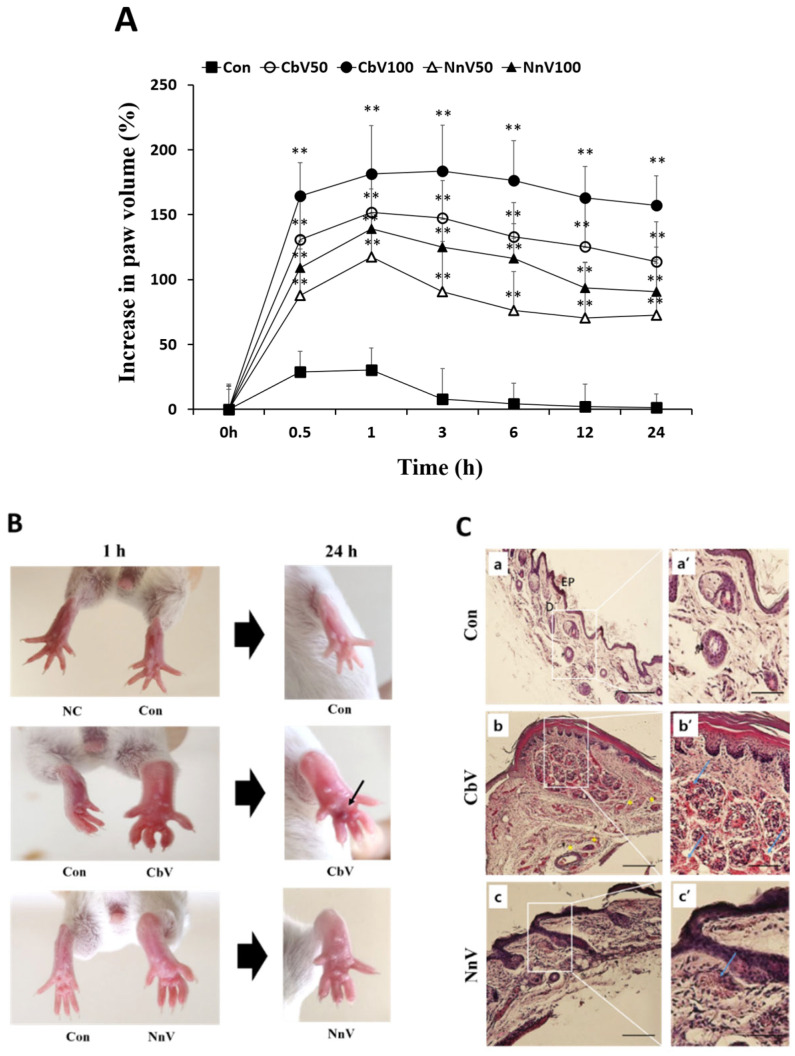
Comparison of the edematogenic activities of CbV and NnV. (**A**) Time course of paw edema development following the injections of either CbV or NnV. BALB/c (*n* = 5) mice were subplantarly injected with either 50 or 100 μg/kg of venom (100 μL) into their right hind paw, with the contralateral hind paw injected with vehicle control (PBS). The edematogenic activity was shown as the edema ratio (%), which is calculated from the percentage increase in the paw thickness using a caliper rule before (T0) and after (Te) the injection with venom or vehicle control. Data represent mean ± SD from 5 mice, ** *p* < 0.01 compared with the vehicle control. (**B**) Photographic images showing mice right hind paw edema after injecting 100 μg of CbV or NnV per mouse at the time point of 1 h and 24 h. CbV injected mice paws showed intense hemorrhagic edemas (a black arrow). (**C**) Histopathological images of mice right hind paws for PBS (control) (**a**,**a’**), CbV (**b**,**b’**), and NnV (**c**,**c’**) treated groups. The local areas of paw tissues framed with the squares in images (**a**–**c**) (scale bars, 100 μm) were magnified and shown in panels (**a’**–**c’**) (scale bars, 50 μm). Ep: Epidermis, D: Dermis. The neutrophil infiltration of paw tissues was shown by blue arrows.

**Figure 5 toxins-14-00831-f005:**
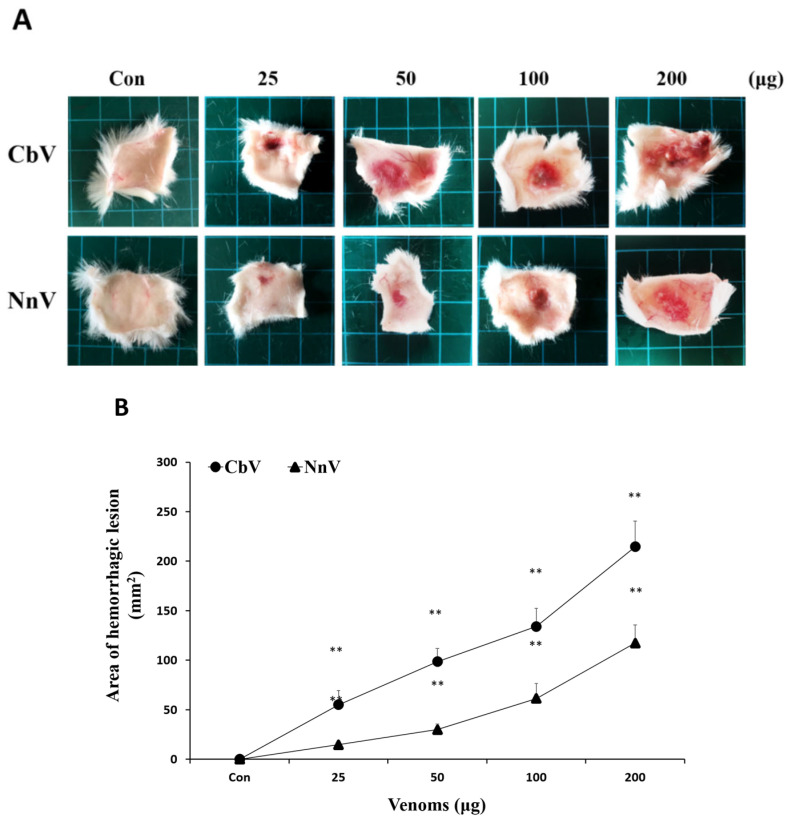
Comparison of the hemorrhagic activities induced by either CbV or NnV. (**A**) Macroscopic view of hemorrhage in mice dorsal skins following the intradermal injection (100 μL) of various amounts (25–200 µg) of CbV or NnV. Control mice were injected with 100 μL of PBS. (**B**) The dose-dependent hemorrhagic effects of CbV and NnV. Quantification of hemorrhagic lesion sizes was presented as the hemorrhagic area (in square millimeters) observed at 3 h after the intradermal venom injection. The results are shown as mean ± S.D. All experiments were performed in triplicate. Data represent mean ± SD from 5 mice, ** *p* < 0.01 compared to the control.

**Figure 6 toxins-14-00831-f006:**
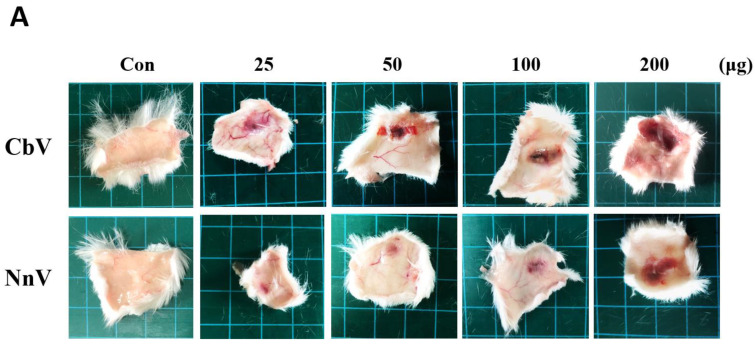
Comparison of the skin necrotic changes induced by either CbV or NnV in mice (after 72 h of intradermal injection). (**A**) Macroscopic view of the internal aspect of mice skin showing dermal necrosis lesion after 72 h of intradermal injection of either CbV or NnV. (**B**) The dose were dependent on dermal necrotic activity induced by either CbV or NnV. Diameters of dermal necrosis lesions were measured for size (mm^2^). The results are expressed as mean ± S.D. All experiments were performed in triplicate. Data represent mean ± SD from 5 mice, * *p* < 0.05, ** *p* < 0.01 compared to the control. (**C**) Histopathological changes of skin sections stained by Hematoxylin eosin, 72 h after intradermal injection with either CbV or NnV. The areas of skin tissues framed with the squares in the images (**a**–**e**) (scale bars, 100 μm) were magnified and shown in panels (**a’**–**e’**) (scale bars, 200 μm). Ep: Epidermis, D: Dermis, Hyp: Hypodermis, Mu: Muscle layer. (**D**) The dermal necrotic lesion was scored layer-by-layer for the dorsal skin intradermally injected with either CbV or NnV. All experiments were performed in triplicate. Data represent mean ± SD from 5 mice, * *p* < 0.05, ** *p* < 0.01 compared with the control. (**E**) TUNEL-like staining of the skin sections 72 h after intradermal injection with either CbV or NnV. The areas of skin tissues framed with the squares in images (**a**–**e**) (scale bars, 100 μm) were magnified and shown in panels (**a’**–**e’**) (scale bars, 200 μm). (**F**) The apoptotic cells were calculated after their quantification in 10 randomly selected fields at 400× magnification under a microscope. Data represent mean ± SD from 5 mice, * *p* < 0.05, ** *p* < 0.01 compared to the control.

**Table 1 toxins-14-00831-t001:** Mortality rates (i.p., 24 hr) of CbV and NnV in mice and their calculated median lethal doses (LD_50_).

Venom (mg/kg)	0	0.25	0.5	1	2.5	5	10	20	Median Lethal Dose (LD_50_)
	Mortality (%)								
CbV	0	0	20	60	80	100	100	100	0.9052
NnV	0	0	0	20	40	60	80	100	4.4697

**Table 2 toxins-14-00831-t002:** Tissue dermal necrosis score.

Necrosis Score	Severity	Description
0	Normal Within	Normal limits
1	Minimal Sporadic	Occurrence
2	Mild Aggregated	Distribution
3	Moderate	Regional distribution
4	Severe Diffuse	Distribution and lose originality

## Data Availability

Not applicable.

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
