# Peer review of "Comparative Study of Toxic Effects and Pathophysiology of Envenomations Induced by Carybdea brevipedalia (Cnidaria: Cubozoa) and Nemopilema nomurai (Cnidaria: Scyphozoa) Jellyfish Venoms"

_toxins, 2022, doi:10.3390/toxins14120831_

Round 1

Reviewer 1 Report

The manuscript “Comparative study of toxic effects and pathophysiology of envenomations induced by Carybdea brevipedalia (Cnidaria:  Cubozoa) and Nemopilema nomurai (Cnidaria: Scyphozoa) jellyfish venoms” is an important study reporting pathophysiological effects of nematocyst aqueous extracts.  This reviewer supports acceptance with minor corrections and changes.

General reviewer comments

It is critical to distinguish a phosphate buffered saline extract of lyophilized nematocysts, which represents a subset of “venom” as it is only comprised of aqueous soluble nematocysts components that survive lyophilization. In contrast authentic venom, i.e.  full live nematocyst content, contains lipidic substances, lipoproteins and other non aqueous soluble components. During an authentic envenomation, the “venom”, that is the full suite of components, injected into prey during a sting is not examined in this manuscript. The authors must clearly state the pitfalls of this approach and the implications that this work will not capture the complete bioactivity or dose range specific activity of full venom that is injected into prey at the time of envenomation. That said, this methodology has been used by other investigators and thus there is a basis for general comparison with other reports in the literature.

The authors overlook key previous peer reviewed publications. The authors claim that this manuscript is the “first” …. In vivo work but omit multiple other examples including:

Yanagihara, A.A., Shohet, R.V. Cubozoan venom-induced acute cardiovascular collapse is caused by hyperkalemia and prevented by zinc gluconate. PLoS One. 2012;7(12):e51368. doi: 10.1371/journal.pone.0051368. Epub 2012 Dec 12. PMCID: PMC3520902

The authors do not cite key examples of porin related- activity, -assays or -pathophysiology:

Yanagihara, A.A., Shohet, R.V. Cubozoan venom-induced acute cardiovascular collapse is caused by hyperkalemia and prevented by zinc gluconate. PLoS One. 2012;7(12):e51368. doi: 10.1371/journal.pone.0051368. Epub 2012 Dec 12. PMCID: PMC3520902

The authors do not cite key literature demonstrating hemolytic activity in hydrozoan and scyphozoans including:

Doyle TK, Headlam JL, Wilcox CL, Macloughlin E, Yanagihara AA, Evaluation of *Cyanea capillata* Sting Management Protocols Using Ex Vivo and In Vitro Envenomation Models, Toxins, 01 July 2017, Vol.9(7), p.215

It would be useful to include heat-treated venom extract results to compare with other published findings. Would 5 min treatment at 100oC  inactivate all effects observed?  This would be an important observation to report. Would 45 min treatment at 45oC of the venom extracts reduce the pathophysiological findings reported here?

 It is suggested that the authors submit the manuscript for English grammatical review and correction. There are many instances of awkward word choices as well as inconsistency in capitalization.

The comments are listed below. Each contains manuscript section or line number, the type of comment/correction,  “original text” in quotations, and suggested texts bracketed by _underscore marks_.

The use of LD50, LD10hr, LD 100 is difficult to follow. Since both time points and percent survival are changing variables in the compound descriptors, it would be clearer to use more detailed abbreviations, for example: 24hr-LD50, 10hr-LD50, 10hr-LD100.

Specific Comments

1. English language. Line 18 “ jellyfish stings result in local skin damages of poisoned tissue and in some 18 cases they cause systemic envenomations “ Jellyfish stings are a type of envenomation so it is not correct to state  “jellyfish stings result in….. envenomations” an alternative could be _jellyfish stings can result in local tissue damage and systemic pathophysiological sequelae_

2. English language. Line 23. Delete the word “ the” before “two”

3. Accuracy. Line 25, 38, 40,44 (Line 100-124). The methods utilized yield a PBS extract of lyophilized nematocyst or an aqueous extract. This is not authentic venom and must be clearly disclosed. “venoms’ should be _aqueous nematocyst extract_

It is accurate to use the term “venom extracts” as the authors have done in lines 120 -121

4. Clarity Line 28. “4.4697/0.905 = 4.939 fold” There is no descriptor of what this is (unlike line 30 where the ratio is clearly explained).

5. English language. Line 48. Delete the word “Nowadays” “have been” and use _are_

6. English language. Line54. Belongs should be _belong_

7. clarity line 57. What does “locomotion” have to do with envenomation?

8. English language. Line58. What are “teguments” ?

9. Accuracy. Line 67. The word “cure” is over stating current management goals.  “Current therapies for curing jellyfish stings, antihistamines were routinely 67 used to attenuate the skin lesions, including rash, itching and oedema [14,15]. “  It would be clearer to state _Current treatment approaches to manage the skin lesions, including rash, itching and oedema include the use of oral and topical antihistamines_

10. Word choice line 87, line 376. “Poisonous” is defined as toxicity to the host organisms after ingestion of an agent or topical application such as ingestion of arsenic or exposure to poison ivy. Venoms are highly specialized toxic compounds and proteins/peptides that are produced by organisms and injected or delivered into the prey through biological means. Rather than “not as poisonous as Cubozoans” another option could be _not as toxic as that of Cubozoans_.

11. Scientific accuracy. Line 91-93. “Currently, to my understanding, there is no standardized measurement to estimate 91 the toxicities of venoms of diverse origins, using objective and quantitative methodologies.”

This comment disregards a very deep literature focused on precisely this topic: biological activities of venoms of diverse origins. Details in general reviewer comments.

12. Technical question. Fig 2 D appears to show far more rod shaped structures than nematocysts. What are these? They do not show internal eversible tubule diagnostic of cnidae. The preparation appears to be heavily contaminated with bacteria or fungi/yeast.

13. Scientific accuracy. Line 236. The authors state “We have, for the first time, compared the lethalities and various toxinological effects 236 of venoms between the two representative Cnidarian classes (C. brevipedalia as a Cubozoa, 237 and N. nomurai as a Scyphozoa“. This statement seems unnecessary and a bit misleading. Other have examine toxinological effects as well as lethal effects. The use of “first” is superfluous and should be removed. The work is important and does not require “first”. The authors should also endeavor to properly review all the pertinent preceding literature. Many key publications have been overlooked or excluded.

14. Scientific accuracy. Line 406-408. ”Additionally, hemolysins are often found in the venoms of vari-406 ous kinds of poisonous jellyfish species, exhibiting the lysis of red blood cells and poten-407 tially causing high lethalities [10,55,56]. The authors again omit pertinent preceding literature. Details in general reviewer comments.

15.  Scientific accuracy. Line 408-415. “Based on previous reports, the box jellyfish ven-408 oms have labile, basic cytolysins (42–46 kDa) that are not known to occur in other Cnidar-409 ians [57-60]. The cytolysins have been demonstrated to be potentially lethal and cause 410 pain, inflammation and necrosis of the skins in experimental animals, as pore-forming 411 toxins [55,57,58]. For example, a box jellyfish Carybdea alata has both hemolytic and lethal activities [58]. Chiropsalmus quadrigatus showed hemolytic activity toward sheep red blood 413 cells with an ED50 of 160 ng/mL and lethality to crayfish with an LD50 of 80 μg/kg [59]. 414 Carybdea rastoni venom was being hemolytic and fatal to mice at 20 μg/kg (i.v) [61]. “ The authors again omit pertinent preceding literature. Details in general reviewer comments.

16. Scientific accuracy. Line 418-420. “Surprisingly, how-418 ever, there were so few attempts to do comparative toxinological study investigating and 419 comparing simultaneously the venoms of cubozoa and scyphozoa.” The authors again omit pertinent preceding literature. Details in general reviewer comments.

Author Response

see formatted content in attachment

General reviewer comments Q1 - It is critical to distinguish a phosphate buffered saline extract of lyophilized nematocysts, which represents a subset of “venom” as it is only comprised of aqueous soluble nematocysts components that survive lyophilization. In contrast authentic venom, i.e. full live nematocyst content, contains lipidic substances, lipoproteins and other non-aqueous soluble components. During an authentic envenomation, the “venom”, that is the full suite of components, injected into prey during a sting is not examined in this manuscript. The authors must clearly state the pitfalls of this approach and the implications that this work will not capture the complete bioactivity or dose range specific activity of full venom that is injected into prey at the time of envenomation. That said, this methodology has been used by other investigators and thus there is a basis for general comparison with other reports in the literature. Response 1: Thank you for the comment. In the present study, we have investigated the toxicities of aqueous extracts of lyophilized nematocyst samples for CbV and NnV. The extraction method is one of the standardized universal methods, and has been widely used for the toxinological studies of jellyfish venoms in the large number of previous research works by other investigators as well. In a point of view, however, we agree with the possible issue raised by the reviewer 1 and included the term “venom extracts” in the revised manuscript. Q2 - The authors overlook key previous peer reviewed publications. The authors claim that this manuscript is the “first” …. In vivo work but omit multiple other examples. Response 2: We appreciate the comment. The aims of our present work is the comparative toxinological evaluation for two representative species of Cubozoa and Scyphozoa, which are currently the most problematic jellyfish classes throughout the world. For this, we have compared the venoms of two jellyfish species side by side using the confirmed experimental methods, especially for the quantified measurement of the venom toxicities, which can be employed for establishing therapeutic strategies against the poisonous stings. Q 3 - The authors do not cite key examples of porin related activity, -assays or -pathophysiology: Response 3: Thank you for the suggestion. In our original manuscript, we have already addressed the porin-like toxin, such as cytolysin, and its presence among the venom components of some of jellyfish species with references. (Line 271-275). We agree with the reviewer’s point that the description of porin related activity, especially for Cubozoan jellyfish venom can be important. Hence, we have cited it in the revision according to the reviewer’s suggestion. (Reference no.51) Q 4 - The authors do not cite key literature demonstrating hemolytic activity in hydrozoan and scyphozoans including: Response 4: We agree with the reviewer’s point and added the article according to the suggestion. (Reference no.50) Q 5 - It would be useful to include heat-treated venom extract results to compare with other published findings. Would 5 min treatment at 100°C inactivate all effects observed? This would be an important observation to report. Would 45 min treatment at 45°C of the venom extracts reduce the pathophysiological findings reported here? Response 5: Previously, we have published the heat sensitivity of N. nomurai jellyfish venom (please refer to the below for details). Kang, C., Munawir, A., Cha, M., Sohn, E. T., Lee, H., Kim, J. S., ... & Kim, E. (2009). Cytotoxicity and hemolytic activity of jellyfish Nemopilema nomurai (Scyphozoa: Rhizostomeae) venom. Comparative Biochemistry and Physiology Part C: Toxicology & Pharmacology, 150(1), 85-90. Q 6 - It is suggested that the authors submit the manuscript for English grammatical review and correction. There are many instances of awkward word choices as well as inconsistency in capitalization. The comments are listed below. Each contains manuscript section or line number, the type of comment/correction, “original text” in quotations, and suggested texts bracketed by _underscore marks_. Response 6: Thank you for the kind suggestions. We have corrected and refined the texts accordingly in the revised manuscript. Q 7 - The use of LD50, LD10hr, LD100 is difficult to follow. Since both time points and percent survival are changing variables in the compound descriptors, it would be clearer to use more detailed abbreviations, for example: 24hr-LD50, 10hr-LD50, 10hr-LD100. Response 7: We appreciate the comment and revised accordingly. Specific Comments Q 1 - English language. Line 18. “ jellyfish stings result in local skin damages of poisoned tissue and in some cases they cause systemic envenomations “ Jellyfish stings are a type of envenomation so it is not correct to state “jellyfish stings result in….. envenomations” an alternative could be _jellyfish stings can result in local tissue damage and systemic pathophysiological sequelae_ Response 1: Thank you for the valuable comments and kind suggestions. We have revised the manuscript accordingly. (Line 20) Q 2 - English language. Line 23. Delete the word “ the” before “two” Response 2: Thank you for your comment. We agree with the point and edit it. (Line 24) Q 3 - Accuracy. Line 25, 38, 40,44 (Line 100-124). The methods utilized yield a PBS extract of lyophilized nematocyst or an aqueous extract. This is not authentic venom and must be clearly disclosed. “venoms’ should be _aqueous nematocyst extract_ It is accurate to use the term “venom extracts” as the authors have done in lines 120 -121 Response 3: We appreciate the comment and revised the manuscript according to the reviewer’s comment. Q 4 - Clarity Line 28. “4.4697/0.905 = 4.939-fold” There is no descriptor of what this is (unlike line 30 where the ratio is clearly explained). Response 4: Thanks for your comments. We agree with the reviewer and rewriting. (Line 31) Q 5 - English language. Line 48. Delete the word “Nowadays” “have been” and use _are_ Response 5: We appreciate the comment and deleted it in the revision. (Line 51) Q 6 - English language. Line54. Belongs should be _belong_ Response 6: Thank you for the precious comment and we edit it in the revision. (Line 57) Q 7 - clarity line 57. What does “locomotion” have to do with envenomation? Response 7: Thanks for your comments. We agree with the reviewer and rewriting. (Line 60) Q 8 - English language. Line58. What are “teguments”? Response 8: Thank you for the comment. We have replaced “tegument” with “integument” which is a little more commonly used for the outer protective layer or covering of an animal, such as skin or cuticle. (Line 61) Q 9 - Accuracy. Line 67. The word “cure” is over stating current management goals. “Current therapies for curing jellyfish stings, antihistamines were routinely 67 used to attenuate the skin lesions, including rash, itching and oedema [14,15]. “ It would be clearer to state _Current treatment approaches to manage the skin lesions, including rash, itching and oedema include the use of oral and topical antihistamines_ Response 9: Thank you for the kind suggestion. We have edited the sentence accordingly. (Line 70) Q 10- Word choice line 87, line 376. Poisonous” is defined as toxicity to the host organisms after ingestion of an agent or topical application such as ingestion of arsenic or exposure to poison ivy. Venoms are highly specialized toxic compounds and proteins/peptides that are produced by organisms and injected or delivered into the prey through biological means. Rather than “not as poisonous as Cubozoans” another option could be _not as toxic as that of Cubozoans_. Response 10: Thank you for the kind suggestion. Although the two terms are often used interchangeably, there can be different opinions on their correct meaning and use between peoples. Hence, we have edited the sentence accordingly. (Line 92, 384) Q 11- Scientific accuracy. Line 91-93. “Currently, to my understanding, there is no standardized measurement to estimate the toxicities of venoms of diverse origins, using objective and quantitative methodologies.” This comment disregards a very deep literature focused on precisely this topic: biological activities of venoms of diverse origins. Details in general reviewer comments. Response 11: The kind comment is greatly appreciated. We have already addressed it in General Reviewer Comments above. (Line 96) Q 12- Technical question. Fig 2 D appears to show far more rod shaped structures than nematocysts. What are these? They do not show internal eversible tubule diagnostic of cnidae. The preparation appears to be heavily contaminated with bacteria or fungi/yeast. Response 12: Thank you for the comment. Nemopilema nomurai tentacle also possess rod shaped nematocysts along with spherical shaped ones. Regarding the point, we have already reported it in our previous report (please refer to the below for details). Pyo MJ, Lee H, Bae SK, Heo Y, Choudhary I, Yoon WD, Kang C, Kim E. Modulation of jellyfish nematocyst discharges and management of human skin stings in Nemopilema nomurai and Carybdea mora. Toxicon. 2016 Jan;109:26-32. doi: 10.1016/j.toxicon.2015.10.019. Epub 2015 Nov 2. PMID: 26541574. Q 13 - Scientific accuracy. Line 236. The authors state “We have, for the first time, compared the lethalities and various toxinological effects of venoms between the two representative Cnidarian classes (C. brevipedalia as a Cubozoa, 237 and N. nomurai as a Scyphozoa“. This statement seems unnecessary and a bit misleading. Other have examine toxinological effects as well as lethal effects. The use of “first” is superfluous and should be removed. The work is important and does not require “first”. The authors should also endeavor to properly review all the pertinent preceding literature. Many key publications have been overlooked or excluded. Responses 13: We have addressed the point in General Reviewer Comments above. Q 14 - Scientific accuracy. Line 406-408. ”Additionally, hemolysins are often found in the venoms of various kinds of poisonous jellyfish species, exhibiting the lysis of red blood cells and potentially causing high lethalities [10,55,56]. The authors again omit pertinent preceding literature. Details in general reviewer comments. Responses 14: We appreciate the comments. We have addressed the point in General Reviewer Comments above. Q 15 - Scientific accuracy. Line 408-415. “Based on previous reports, the box jellyfish venoms have labile, basic cytolysins (42–46 kDa) that are not known to occur in other Cnidarians [57-60]. The cytolysins have been demonstrated to be potentially lethal and cause pain, inflammation and necrosis of the skins in experimental animals, as pore-forming toxins [55,57,58]. For example, a box jellyfish Carybdea alata has both hemolytic and lethal activities [58]. Chiropsalmus quadrigatus showed hemolytic activity toward sheep red blood cells with an ED50 of 160 ng/mL and lethality to crayfish with an LD50 of 80 μg/kg [59]. Carybdea rastoni venom was being hemolytic and fatal to mice at 20 μg/kg (i.v) [61]. “ The authors again omit pertinent preceding literature. Details in general reviewer comments. Responses 15: We appreciate the comments. We have addressed the point in General Reviewer Comments above. Q 16 - Scientific accuracy. Line 418-420. “Surprisingly, however, there were so few attempts to do comparative toxinological study investigating and comparing simultaneously the venoms of cubozoa and scyphozoa.” The authors again omit pertinent preceding literature. Details in general reviewer comments. Responses 16: We appreciate the comments. We have addressed the point in General Reviewer Comments above.

Reviewer 2 Report

The study entitled Comparative study of toxic effects and pathophysiology of envenomations induced by Carybdea brevipedalia (Cnidaria: Cubozoa) and Nemopilema nomurai (Cnidaria: Scyphozoa) jellyfish venoms submitted to Toxins by Du Hyeon Hwang and co-workers have examined and compared the in vivo toxic effects and pathophysiologic alterations using experimental animal models for the two representative stinging jellyfish classes, i.e., Cubozoa and Scyphozoa. The manuscript focuses an interesting topic that is worth to be published, but after minor revision.

-Moderate English changes are required

-In general, I suggest to review the style of the manuscript according to the guidelines of the journal.

-In the abstract delete In general, start the paragraph with Jellyfish stings.

-In the first part of introduction the authors could be introduce like comparison the effect of different agents upon the haemolytic power of crude venom in human erythrocytes to determine its toxicity and stability. For these reasons you can considered followed articles 10.1016/j.cbpa.2008.11.016 ; https://doi.org/10.2174/1871524914666141028150212

-Line 381: These topical cutaneous symptoms are often inflammatory, and termed as jellyfish contact dermatitis. Thus, I suggest to add the following references https://doi.org/10.3390/md12042182

Author Response

see the formatted content in attachment

Comments and Suggestions for Authors The study entitled “Comparative study of toxic effects and pathophysiology of envenomations induced by Carybdea brevipedalia (Cnidaria: Cubozoa) and Nemopilema nomurai (Cnidaria: Scyphozoa) jellyfish venoms “submitted to Toxins by Du Hyeon Hwang and co-workers have examined and compared the in vivo toxic effects and pathophysiologic alterations using experimental animal models for the two representative stinging jellyfish classes, i.e., Cubozoa and Scyphozoa. The manuscript focuses an interesting topic that is worth to be published, but after minor revision. Response: Thank you for the kind comment. Q1 - Moderate English changes are required. Response 1: We appreciate the comment and revised English in the revised manuscript accordingly. Q2 - In general, I suggest to review the style of the manuscript according to the guidelines of the journal. Response 2: Thank you for the precious comment. We have carefully changed our manuscript according to the ‘Toxins’ journal style based on the guidelines. Q3 - In the abstract delete In general, start the paragraph with Jellyfish stings. Response 3: Thank you for your comment. We agree with the point and have deleted the word ‘In general,’ in the abstract of the revised one. Q4 - In the first part of introduction the authors could be introduce like comparison the effect of different agents upon the hemolytic power of crude venom in human erythrocytes to determine its toxicity and stability. For these reasons you can considered followed articles 10.1016/j.cbpa.2008.11.016 ; https://doi.org/10.2174/1871524914666141028150212 Response 4: We appreciate the comment. Regarding the point, we plan to submit for another manuscript in the near future. Q5 - Line 381: These topical cutaneous symptoms are often inflammatory, and termed as jellyfish contact dermatitis. Thus, I suggest to add the following references https://doi.org/10.3390/md12042182. Response 5: We appreciate the comment. We agree with the point, and have included it in the reference. (reference no. 33) (Bruschetta, G., Impellizzeri, D., Morabito, R., Marino, A., Ahmad, A., Spanò, N., ... & Esposito, E. (2014). Pelagia noctiluca (Scyphozoa) crude venom injection elicits oxidative stress and inflammatory response in rats. Marine Drugs, 12(4), 2182-2204.)

Round 2

Reviewer 1 Report

The authors have made strong progress towards addressing grammatical and scientific literature recitation issues raised by this reviewer to the prior submission. There are remaining grammatical issues.

All remaining issues are highlighted in an attached file with "sticky note" reviewer comments. 

To summarize the critical specific issues that remain: the annotation of LD measurements remains a bit confusing in places. Is "10hr-LD" 10hr-LD50 or 10hr-LD100Venom components have been shown to be highly conserved in that all cnidarians studied to date contain similar classes of enzymes and toxins. That said, the specific amino acid sequence and specific activity clearly varies. Thus, while all transcriptomic and biochemical studies show that cnidarian venoms contain representative proteases, lipases, pore forming proteins and small peptide toxins, the specific activity of each has been found to vary from species to species. This topic has been discussed in greater detail in the "sticky notes".  Finally, the term "envenomation" is a medically and scientifically accurate description of the active deposition by way of a bite or other mechanism of a venom (biological mixture of toxins produced by an organism) into prey or victim. The term "sting" is a lay term which is synonymous with envenomation. For this reason it is not accurate to state that a "sting... causes... envenomation". Alternatives have been suggested in "sticky note" comments by the reviewer. 

Author Response

Responses to Reviewer 1 Comments

Comments and Suggestions for Authors

Q - The authors have made strong progress towards addressing grammatical and scientific literature recitation issues raised by this reviewer to the prior submission. There are remaining grammatical issues.

All remaining issues are highlighted in an attached file with "sticky note" reviewer comments. 

To summarize the critical specific issues that remain: the annotation of LD measurements remains a bit confusing in places. Is "10hr-LD" 10hr-LD50 or 10hr-LD100?  

Venom components have been shown to be highly conserved in that all cnidarians studied to date contain similar classes of enzymes and toxins. That said, the specific amino acid sequence and specific activity clearly varies. Thus, while all transcriptomic and biochemical studies show that cnidarian venoms contain representative proteases, lipases, pore forming proteins and small peptide toxins, the specific activity of each has been found to vary from species to species. This topic has been discussed in greater detail in the "sticky notes".  Finally, the term "envenomation" is a medically and scientifically accurate description of the active deposition by way of a bite or other mechanism of a venom (biological mixture of toxins produced by an organism) into prey or victim. The term "sting" is a lay term which is synonymous with envenomation. For this reason it is not accurate to state that a "sting... causes... envenomation". Alternatives have been suggested in "sticky note" comments by the reviewer. 

Response : Thank you for the comments and suggestions.

We have revised our manuscript according to the comments and suggestions.